# Polygenic Adaptation in a Population of Finite Size

**DOI:** 10.3390/e22080907

**Published:** 2020-08-18

**Authors:** Wolfgang Stephan, Sona John

**Affiliations:** 1Leibniz Institute for Evolution and Biodiversity Science, Natural History Museum, 10115 Berlin, Germany; stephan@bio.lmu.de; 2Department of Life Science Systems, Technical University of Munich, 85354 Freising, Germany

**Keywords:** rapid adaptation, highly polygenic trait, population genetics, genetic drift

## Abstract

Polygenic adaptation in response to selection on quantitative traits has become an important topic in evolutionary biology. Here we review the recent literature on models of polygenic adaptation. In particular, we focus on a model that includes mutation and both directional and stabilizing selection on a highly polygenic trait in a population of finite size (thus experiencing random genetic drift). Assuming that a sudden environmental shift of the fitness optimum occurs while the population is in a stochastic equilibrium, we analyze the adaptation of the trait to the new optimum. When the shift is not too large relative to the equilibrium genetic variance and this variance is determined by loci with mostly small effects, the approach of the mean phenotype to the optimum can be approximated by a rapid exponential process (whose rate is proportional to the genetic variance). During this rapid phase the underlying changes to allele frequencies, however, may depend strongly on genetic drift. While trait-increasing alleles with intermediate equilibrium frequencies are dominated by selection and contribute positively to changes of the trait mean (i.e., are aligned with the direction of the optimum shift), alleles with low or high equilibrium frequencies show more of a random dynamics, which is expected when drift is dominating. A strong effect of drift is also predicted for population size bottlenecks. Our simulations show that the presence of a bottleneck results in a larger deviation of the population mean of the trait from the fitness optimum, which suggests that more loci experience the influence of drift.

## 1. Introduction

Evolutionary adaptation is the process that natural populations experience to become suited to their environment. Adaptation is driven by positive Darwinian selection, which leaves footprints in the DNA of individuals. For the past almost 20 years numerous methods have been developed to find evidence for positive selection in the genomes of natural populations [1,2,3,4]. The population genetic models underlying these methods consider mostly single loci at which positive selection acts, in particular in the context of genetic hitchhiking [5,6,7]. Occasionally, however, extensions of single-locus models to multi-locus models were also investigated [8,9,10]. In recent years, due to the advance of genome-wide association studies (GWAS), polygenic selection was studied using quantitative genetics models that are formulated in terms of allele frequency changes in a large number of loci across the whole genome. That is, in contrast to the single- and multi-locus cases, in polygenic models, selection acts on a phenotypic trait, and a genotype-phenotype map is assumed to bridge the gap to population genetics. Polygenic models include a very large number of selected loci that control a phenotype [11].

For any type of adaptive scenario, positive selection has to be combined with other factors characterizing the status of a natural population, to make the models more realistic and applicable to data. These complexities include genetic drift, which takes into account that populations have a finite number of individuals. Furthermore, demographic changes such as varying population size may be relevant, and positive selection in combination with population structure needs to be modeled to describe typical phenomena, such as local adaptation, which has been an important topic in population genetics in recent years [12,13,14,15].

Polygenic adaptation driven by a large number of weakly selected loci is not nearly as well studied as the case of strong positive selection leading to selective sweeps [16]. Reviews by Pritchard et al. (2010) [17] and Pritchard and Di Rienzo (2010) [18] drew the attention of population geneticists to this type of selection. These papers predicted that allele frequencies change by small amounts when a large number of genetic loci of minor effect sizes control a phenotypic trait. However, it is not obvious whether polygenic adaptation may be so fast, as suggested by an increasing number of cases reported in the recent literature.

Examples of very rapid adaptation in response to changes that are natural or due to human activity include color variation in guppies [19], field mice [20] and peppered moth [21]; insecticide resistance in Drosophila [22]; beak size changes in Darwin’s finches [23], and limb development in Anolis lizards [24]. The genetic architecture underlying these phenotypic traits ranges from few genes of major effect as in the peppered moth [25] to highly polygenic systems such as human height [26].

The analysis of models of polygenic adaptation has a relatively long tradition [27,28]. However, it is beyond the scope of this paper to provide a comprehensive overview of the history of this field. Instead we review the recent literature on theoretical advances on polygenic adaptation, and then focus on a particular model along with few new results. De Vladar and Barton (2014) [29] and Jain and Stephan (2015) [30] used a deterministic model to analyze the dynamics of adaptation after a sudden environmental shift of the fitness optimum of a phenotypic trait in the absense of genetic drift. According to Jain and Stephan (2017a, b) [31,32], this model predicts that rapid adaptation may occur either through strong directional selection at a few loci (when the effect sizes of the alleles at these loci are large relative to a scaled mutation rate), through weak selection at many loci with small effect sizes or through a combination of these two modes. In the case of many loci with small effects subtle allele frequency shifts are typical. However, all these studies assumed infinitely large population sizes. This led to problems in defining the values of the allele frequencies at the time of the environmental change, as the equilibrium allele frequencies of the deterministic model do not agree with the frequencies typically observed in GWAS.

Recognizing this problem, the studies published in the past few years [33,34,35,36,37] include genetic drift (due to finite population size) into their polygenic models. Simons et al. (2018) [35] proposed a model of selection that simultaneously acts on multiple traits (pleiotropy). For a single trait their model is identical to the one we discuss below. Stetter et al. (2018) [36] and Thornton (2019) [37] used extensive forward simulations to analyze a model (though with relatively few selected loci) that also includes neutral loci linked to selected ones. This enables them to study genetic hitchhiking in a (potentially) polygenic context. Stetter et al. (2018) [36] also investigated the effect of different demographies and genetic architectures of the trait. The model of Höllinger et al. (2019) [33] is different from ours in that the loci controlling a trait are not explicitly given, but instead a genome-wide mutation rate is used as a proxy. Despite this difference, their main conclusions on the modes of rapid adaptation are very similar to our predictions (see previous paragraph).

Here we first review the work of John and Stephan (2020) [34] in which we described a stochastic treatment of the equilibrium phase before the shift of the fitness optimum. This led to a reasonable definition of allele frequencies at the timepoint of the environmental change. Second, we present a new analysis of adaptation in response to selection in the rapid short-term phase after the optimum shift based on diffusion theory. Finally, of the additional factors influencing a natural population (listed above), we study the effect of varying population size by computer simulation.

## 2. Model and Analysis

### 2.1. Deterministic Model of a Single Quantitative Trait

We begin by describing briefly the model of a quantitative trait that has been used by de Vladar and Barton (2014) [29], Jain and Stephan (2015, 2017a, b) [30,31,32], and John and Stephan (2020) [34]. We consider a perfectly heritable trait that is controlled additively (no dominance or epistasis) by *l* unlinked, diallelic loci in a very large population of diploids. If the phenotypic effect of the trait-increasing allele, called + allele, at locus *i* is $\frac{\gamma_{i}\;}{2}$ and that of the – allele is $\frac{-\gamma_{i}\;}{2}$, the mean phenotype $c_{1}$, the genetic variance $c_{2}$ and the skewness $c_{3}$ are given by [31]
(1)$$c_{1}=\sum_{i=1}^{l}\gamma_{i}\left(p_{i}-q_{i}\right)=\sum_{i=1}^{l}\gamma_{i}\left(2p_{i}-1\right)\;,$$
(2)$$c_{2}=2\sum_{i=1}^{l}\gamma_{i}^{2}p_{i}q_{i\;\;},\;\mathrm{and}\;$$
(3)$$c_{3}=2\sum_{i=1}^{l}\gamma_{i}^{3}p_{i}q_{i\;\;}\left(q_{i}-p_{i}\right)\;,\;$$
where $p_{i}$ is the frequency of the + allele at locus *i* and $q_{i}=1-p_{i}$ is that of the – allele. We assume that the effect sizes are exponentially distributed with mean $\bar{\gamma }.$ Furthermore, the fitness of an individual with trait value z has a Gaussian shape centered about the fitness optimum $z_{0}$
(4)$$w\left(z\right)=e^{-\frac{s}{2}{\left(z-z_{0}\right)}^{2}}\;,$$
where *s* measures the strength of selection on the trait. 1/*s* is assumed to be much larger than the phenotypic variance [35]. Without loss of generality, we also assume $0<z_{0},$ and require that $z_{0}$
$<l\bar{\gamma }.$ The latter condition ensures that the population mean converges to a stationary state close to the optimum [30]. In a randomly mating population, the change in allele frequency at the *i*th locus due to selection and mutation is then given by
(5)$$\frac{dp_{i}}{dt}=-s\gamma_{i}p_{i}q_{i}\Delta c_{1}-\frac{s\gamma_{i}^{2}}{2}p_{i}q_{i}\left(q_{i}-p_{i}\right)-\mu p_{i}+\nu q_{i}\;,\;\;i=1,\ldots ,\;l,\;$$
where $\Delta c_{1}=c_{1}-z_{0}$ is the deviation of the mean phenotype from the fitness optimum. The first term on the right-hand side of Equations (5) describes directional selection toward the phenotypic optimum, the second term accounts for stabilizing selection in the vicinity of the optimum [38], and the last two terms for mutation [39,40]. Here, μ represents the mutation rate from the + to the – allele and ν that of the – to the + allele. In agreement with the latter authors, we assume equal forward and backward mutation rates μ=ν in our analysis of this model.

### 2.2. Stochastic Analysis

To apply our polygenic model to a population of finite size *N*, we first analyze it under equilibrium conditions. Then, while the population is in equilibrium, we introduce an optimum shift from $z_{0}$ to $z_{f}$, and allow the population to adapt to the new optimum. In both cases analytical results are obtained based on standard diffusion theory [41]. At the end of this paper, we study the effect of demography (population size changes) on polygenic adaptation, using computer simulations. The simulation procedure is described in John and Stephan (2020) [34].

## 3. Results

We consider the following scenario. At equilibrium, the population mean fluctuates around a state close to the fitness optimum $z_{0}$, until the optimum is shifted suddenly to a new value $z_{f}$. While the analysis of the equilibrium phase is similar to the treatment in John and Stephan (2020) [34], a new approach for the adaptive phase is presented.

### 3.1. Stochastic Equilibrium between Drift, Mutation and Selection

In the polygenic case most of the loci are assumed to have small effects such that $\gamma_{i}$ < $\hat{\gamma },$ where $\hat{\gamma }$
$=2\sqrt{\frac{2\mu }{s}\;}$ [29]. To analyze the equilibrium fluctuations mentioned above, we recall that in the deterministic case the trait mean may change much faster after a perturbation than the allele frequencies [30]. Therefore, it is convenient to express Equations (5) as follows:(6)$$\frac{d\Delta c_{1}}{dt}=-\frac{s}{2}c_{3}-sc_{2}\Delta c_{1}-2\mu c_{1}$$
and
(7)$$\frac{dp_{i}}{dt}=-s\gamma_{i}p_{i}q_{i}\Delta c_{1}-\frac{s\gamma_{i}^{2}}{2}p_{i}q_{i}\left(q_{i}-p_{i}\right)+\mu (q_{i}-p_{i}),\;\;i=1,\ldots ,\;l-1.\;$$

Equation (6) is derived by summing over the Equations (5) and using the definitions of the cumulants (Equations (1)–(3)). Assuming that $\Delta c_{1}$ is a fast variable on the time scale of the allele frequencies $p_{i}$ (Gardiner 1990 [42], chapt. 6.4), we obtain $\Delta {\tilde{c}}_{1}$ by putting the left-hand side of Equation (6) to zero. Then, in quasi-equilibrium the deviation of the population mean from $z_{0}$ is approximately
(8)$$\Delta {\tilde{c}}_{1}\approx -\frac{\frac{s}{2}{\tilde{c}}_{3}+2\mu z_{0}}{s{\tilde{c}}_{2}+2\mu }\;,$$
where tilde indicates the quasi-equilibria of the cumulants involved. The variance is relatively constant [30]. The skewness term varies with time but may be neglected if we assume that the effect sizes are very small (see Equation (3)). The latter assumption has been made by John and Stephan (2020) [34] (see their Equation (8)).

Equation (8) predicts that in equilibrium the trait mean does not converge completely to the fitness optimum. In this model the allele frequencies of Equations (5) approach stable equilibrium states that are incompatible with the fitness optimum (see de Vladar & Barton 2014 [29], Figure 2 and Appendix B).

The expected change of the allele frequency $p_{i}$ may be approximated as
(9)$$E\left\{\Delta p_{i}\right\}\approx -s\gamma_{i}p_{i}q_{i}\Delta {\tilde{c}}_{1}-\frac{s\gamma_{i}^{2}}{2}p_{i}q_{i}\left(q_{i}-p_{i}\right)+\mu (q_{i}-p_{i})\;,$$
where $\Delta {\tilde{c}}_{1}$ is relatively constant. Furthermore, the variance of the change in $p_{i}$ accounting for the effect of drift is
(10)$$Var\left\{\Delta p_{i}\right\}\approx \frac{p_{i}q_{i}}{2N}\;.\;$$

Using diffusion theory (Ewens 2004 [41], chapt. 4.5), this leads to the equilibrium frequency distribution of the trait-increasing allele $p_{i}$ at locus *i*:(11)$$f\left(p_{i}\right)\approx Cp_{i}^{2\beta -1}q_{i}^{2\beta -1}exp\left(-2\alpha \gamma_{i}\Delta {\tilde{c}}_{1}p_{i}-\alpha \gamma_{i}^{2}p_{i}q_{i}\right)\;,\;$$
where *C* is the normalization constant (omitting index *i* for locus *i*), α=2Ns, and β=2Nμ is the scaled mutation rate. *C* is approximately given by [34]
(12)$$C^{-1}\approx B\left(2\beta ,2\beta \right)\left[1-\alpha \gamma_{i}\Delta {\tilde{c}}_{1}-\alpha \gamma_{i}^{2}\frac{\beta }{4\beta +1}\right]\;,$$
where *B* denotes the beta function.

We compared the theoretical results with simulations using the following set of parameter values: *s* = 0.1, *N* = $2\times 10^{4}$, *l*
= 200, *μ* =$10^{-5},\;\bar{\gamma }=0.01$ and $z_{0}$ = 0.2. These values were taken from the literature on polygenic adaptation in humans [26,35]. The stationary mean deviation is slightly negative and agrees reasonably well with the value predicted by Equation (8). Our simulations of the demographic model yielded in equilibrium (before the bottleneck; see Section 3.3) $\Delta {\tilde{c}}_{1}\;$= − 0.0022, ${\tilde{c}}_{2}$ = 0.0089 and ${\tilde{c}}_{3}$ = −5$\times 10^{-5}$, such that Equation (8) predicts $\Delta {\tilde{c}}_{1}=-$0.0016.

### 3.2. Adaptation after a Sudden Shift of the Fitness Optimum

We assume that a population is in equilibrium when the fitness optimum $z_{0}$ is suddenly shifted to a new value $z_{f}>z_{0}$, where $z_{f}<l\bar{\gamma }$. We analyze the dynamics of the alleles at all loci until the population has adapted to the new optimum, i.e., until the population mean has reached a value close to $z_{f}$. Since the trait mean responds rapidly to the shift of the fitness optimum, it may be approximated by an exponential process whose rate is proportional to the equilibrium genetic variance [30]:(13)$$\Delta c_{1}\left(t\right)\approx \Delta c_{1}\left(0\right)exp\left(-sc_{2}\left(0\right)t\right)\;,$$
where $\Delta c_{1}\left(0\right)=c_{1}\left(0\right)-z_{f\;}.$ The variable *t* is measured in generations such that *t* = 0 is the timepoint when the fitness optimum shifts to its new value.

Lande (1976) [28] was first in deriving Equation (13) assuming that the genetic variance is constant (see Equations (17a) and (18a) in Lande (1976) [28]). Numerical simulations with different population sizes show a change in the speed of the approach to the optimum (Figure 1), where $c_{2}\left(0\right)$ accounts for the drift effect in the equilibrium phase (Equation (10)) and $\Delta c_{1}\left(0\right)$ is nearly independent of population size (as $\Delta {\tilde{c}}_{1}$ is close to zero).

Equation (13) characterizes the short-term phase of polygenic adaptation [31]. The short-term phase is the time until the phenotypic mean reaches a value close to the new optimum. It lasts about ${\left(sc_{2}\left(0\right)\right)}^{-1}$ generations. Thus, it may be very short when the variance is large. According to Equation (14) of John and Stephan (2020) [34], this is the case when the number of loci controlling the trait is large and/or the scaled mutation rate *β* is not too small. Indeed, genetic variance may be much reduced below the deterministic value of $l{\bar{\gamma }}^{2}$ when the mutation parameter is small such that the distribution (11) is extremely U-shaped.

Next we describe the stochastic changes of the frequencies of the trait-increasing alleles $p_{i}$ using a diffusion approximation (similar to the approach of Stephan et al. (1992) [7]). The differential operator $L_{i}$ of the Kolmogorov backward equation is
(14)$$L_{i}=\left(-s\gamma_{i}p_{i}\left(1-p_{i}\right)\Delta c_{1}\left(t\right)\right)\frac{\partial }{\partial p_{i}}+\frac{p_{i}\left(1-p_{i}\right)}{4N}\frac{\partial^{2}}{\partial p_{i}^{2}}\;.$$

In contrast to Equation (5), the drift term of this operator includes only the effect of directional selection; i.e., in the rapid initial phase when the population is far from the new fitness optimum stabilizing selection and mutation are much less important than directional selection [30,31]. Furthermore, note that this drift term contains the time-dependent function $\Delta c_{1}\left(t\right)$ defined in Equation (13). To derive equations for the lowest-order moments of the allele frequencies we use the equation (Ewens 2004 [41], chapt. 10.4)
(15)$$\frac{d}{dt}E\left\{f,\;t\right\}=E\left\{L\left(f\right),\;t\right\}\;,\;$$
where L is an appropriately defined differential operator (see above) and f a twice differentiable function. Inserting $p_{i}$ and $p_{i}^{2}$ into Equation (15) for the function f leads to the following ordinary differential equations (ODEs)
(16)$$\frac{d}{dt}E\left\{p_{i}\right\}=-s\gamma_{i}E\left\{p_{i}\left(1-p_{i}\right)\right\}\Delta c_{1}\left(t\right)\;,$$
(17)$$\frac{d}{dt}E\left\{p_{i}^{2}\right\}=-2s\gamma_{i}E\left\{p_{i}^{2}\left(1-p_{i}\right)\right\}\Delta c_{1}\left(t\right)+\frac{1}{2N}E\left\{p_{i}\left(1-p_{i}\right)\right\}.$$

Combining these two ODEs yields
(18)$$\frac{d}{dt}E\left\{p_{i}\left(1-p_{i}\right)\right\}=-s\gamma_{i}E\left\{p_{i}\left(1-p_{i}\right)\right\}\Delta c_{1}\left(t\right)+2s\gamma_{i}E\left\{p_{i}^{2}\left(1-p_{i}\right)\right\}\Delta c_{1}\left(t\right)-\frac{1}{2N}E\left\{p_{i}\left(1-p_{i}\right)\right\}\;.$$

The system of Equations (16) and (18) cannot be solved as the moment expansion does not break up. Nonetheless, Equation (18), in combination with Equation (2), suggests that the contribution of individual loci to genetic variance decreases with increasing genetic drift (i.e., lower population size). Numerical simulations comparing the allele frequency changes at loci with similar effect sizes support this expectation. We can clearly see that the smallest population shows only a minor directional change in allele frequency indicating a stronger impact of genetic drift (Figure 2).

We used simulations to explore the effect of genetic drift on allele frequencies in more detail. Of particular interest are the frequency shifts $\delta p_{i}$ of the alleles during the short-term phase (defined above). Since we assume that $z_{f}>z_{0}$ and thus $\Delta c_{1}\left(0\right)<0$, the allele frequencies $p_{i}\left(t\right)$ are expected to increase with time (see Equation (16)). In the deterministic case, the allele frequency shifts at the end of the short-term phase (i.e., after ${\left(sc_{2}\left(0\right)\right)}^{-1}$ generations) for sufficiently small effect sizes are approximately [34]
(19)$$\delta p_{i}\approx -\gamma_{i}p_{i}\left(0\right)q_{i}\left(0\right)\Delta c_{1}\left(0\right)\frac{1-e^{-1}}{c_{2}\left(0\right)}\;.\;$$

Equation (19) suggests that the allele frequency shift at a locus depends strongly on the compound parameter $\gamma_{i}p_{i}\left(0\right)q_{i}\left(0\right).$ This is similar to the one-locus case, where the product of p and q determines the speed of allele frequency change. The compound parameter increases with the effect size and is greatest for initial frequencies that are intermediate. Furthermore, Equation (19) predicts that after an environmental change the allele frequencies shift coherently into the same direction. This is an important property of polygenic selection because it may help detecting this type of selection, although the frequency shifts at individual loci are in general small.

Including genetic drift, however, leads to a more complex picture of polygenic adaptation. As shown in Figure 1, we find a good agreement between Equation (13) and the simulation for the deviation $\Delta c_{1}$ of the population mean from the optimum within the short-term phase. For the allele frequencies, however, we get a reasonable agreement of the deterministic prediction of Equations (19) and simulations only when the effect sizes are sufficiently large and allele frequencies at the time of the environmental shift are intermediate. As revealed by Equations (16) and (18), the reason is that genetic drift slows down the increase of the allele frequencies and hence reduces the expected differences between the allele frequencies at the end of the short-term phase and those at *t* = 0. As a consequence, while trait-increasing alleles with intermediately high equilibrium frequencies contribute positively to changes of the trait mean (i.e., are aligned with the direction of the optimum shift), alleles with low or high frequencies may not stay aligned with the optimum shift.

### 3.3. Effects of Demography on Polygenic Adaptation

We simulated a simple demographic model with a major bottleneck and subsequent recovery (as inferred from human polymorphism data; Schiffels & Durbin 2014 [43]). The question we ask in this section is to what extent genetic variance, which is a determinant of the speed of adaptation of a polygenic trait (see Equation (13)), is affected by demography.

We started the simulations in the distant past with a population size $N=2\times 10^{4}$. N is assumed to be constant for several thousand generations (such that the population reached an approximate equilibrium state) before it decreased instantaneously to a much lower number of individuals (Figure 3a). The population then stayed at this bottleneck size (of 3000 individuals) for 5000 generations before it instantaneously changed back to the constant size of $2\times 10^{4}$. 100 generations ago population size increased instantaneously to $6\times 10^{5}$. This recent recovery phase was not considered by John and Stephan (2020) [34].

We obtained the following results (Figure 3b). In the pre-bottleneck phase $c_{1}\;$is close to the fitness optimum $z_{0}$ = 0.2, such that $\Delta c_{1}$ is slightly negative (−0.0022). During the bottleneck $c_{1}$ decreases to an average value such that $\Delta c_{1}$ is considerably more negative (−0.0031) than before the bottleneck. This amounts to an approximate 41% increase in the absolute size of $\Delta c_{1}$ after the bottleneck. In the third phase, after population size recovered to $\;2\times 10^{4},$
$c_{1}$ remains lower than at the beginning of the bottleneck. Thus, due to the bottleneck effect the population mean of the trait deviates from the fitness optimum more than before the bottleneck. This observation is caused by genetic drift. Indeed, drift reduces the genetic variance. It drops from 0.0089 to 0.0051 during the bottleneck, which corresponds to a decrease of about 43% at the end of the bottleneck phase (relative to its value at the beginning of the bottleneck) and may thus have a considerable effect on the speed of adaptation (Figure 3b). This is qualitatively in agreement with Equation (8). The dramatic increase of population size in the last 100 generations had almost no effect on the trait mean.

## 4. Discussion

### 4.1. Summary

We analyzed polygenic adaptation in a finite population experiencing a sudden environmental change (while in equilibrium). When the shift of the fitness optimum is not too large relative to the genetic variance in the trait and the variance is mostly due to loci with small effects, we found that the new optimum is approached exponentially at a rate proportional to the equilibrium genetic variance before the optimum shift. This result agrees with Lande’s (1976) [28] prediction of the response of a phenotypic trait to selection, which was obtained based on the infinitesimal model (i.e., a model with an infinite number of loci with infinitesimally small effects). In contrast to the infinitesimal model, our results were derived from a model that contains a large, but finite number of loci with finite effect sizes.

We also analyzed the underlying allelic dynamics of the phenotypic response to selection. Whereas in a population of infinite size, the equilibrium frequencies of small-effect alleles are intermediate [29], we found that genetic drift (in conjunction with symmetric mutation and selection) may lead to a very different equilibrium distribution of allele frequencies. In relatively small populations and for realistic mutation rates, this distribution is U-shaped. This has important consequences for the allelic response to selection. Assuming that a sudden environmental shift of the fitness optimum occurs while the population is in equilibrium, we studied the adaptation of the trait to the new optimum in the short-term phase (which is defined by ${\left(sc_{2}\left(0\right)\right)}^{-1}$ generations). Only alleles with intermediately high equilibrium frequencies contribute positively to changes of the trait mean (i.e., are aligned with the optimum shift). In contrast, alleles with very low or high frequencies are subject to stronger drift and thus may not stay aligned with the direction of the optimum shift.

The effect of genetic drift is also observed in population size bottlenecks. Simulating a bottleneck with subsequent recovery we found that the phenotypic mean of a quantitative trait decreased with the decreasing genetic variance during the bottleneck and was at the end of the bottleneck further way from the fitness optimum (than before the bottleneck).

### 4.2. Impact of Large-Effect Alleles

All the aforementioned results hold for the case when the shift of the fitness optimum is not too large relative to the genetic variance in the trait and the variance is mostly due to loci with small effects. In a very recent study, Hayward and Sella (2020) [44] have generalized some of these findings. They found that when the environmental shift is large relative to the phenotypic standard deviation or when loci of large effect contribute substantially to genetic variance, the initial exponential change of the mean phenotype is similar as in the case of mostly small-effect loci. However, in contrast to the small-effect case, this rapid phase is then followed by a long equilibration period determined by the skewness of the phenotypic distribution, in which the mean phenotype approaches the new optimum.

When large-effect alleles play a substantial role, the allelic response to an environmental shift in the fitness optimum is somewhat different from the case of mostly small-effect alleles [31]. Hayward and Sella (2020) [44] have investigated these differences in detail. Since an allele’s contribution to phenotypic change is proportional to its contribution to phenotypic variance before the shift, alleles with moderate or large effect sizes make the greatest contributions to phenotypic change. In the prolonged equilibration period, stabilizing selection on the trait transforms the small frequency differences that have arisen through directional selection during the rapid initial phase between alleles aligned with the shift and those opposed into a small excess of fixed aligned alleles relative to opposing ones. This excess drives the population mean phenotype all the way to the new optimum. Thereby, the contributions of large-effect alleles are dominated by the effects of fixed moderate-size alleles.

### 4.3. Do Selective Sweeps Occur in Polygenic Adaptation?

This question has been addressed by several authors [31,45,46,47]. Using the deterministic model of Equations (5) Jain and Stephan (2017a) [31] found that selective sweeps may arise in restricted parameter ranges, but only when most alleles have large effects. Indeed, strong selective fixations have been observed in simulations in the initial rapid phase (see Figure 3 of Jain & Stephan 2017a [31]). Furthermore, fixations driven by relatively weak selection may occur in the prolonged equilibration period, but these would not lead to sweeps (i.e., substantial drops of local genetic diversity in the neighborhood of selected sites). These findings agree with the study of Hayward and Sella (2020) [44]. They observed that in highly polygenic models large-effect alleles almost never sweep to fixation, while alleles of moderate effects may go to fixation (see above). In general, in quantitative genetics models, selective sweeps are rare [45,46,47,48]. However, this does not contradict Thornton’s (2019) [37] observation of sweeps in cases in which the trait is not highly polygenic. This was also found by Jain and Stephan (2017a) [31] when the number of loci controlling a trait was not large (see e.g., their Figure 3).

## Figures and Tables

**Figure 1 entropy-22-00907-f001:**
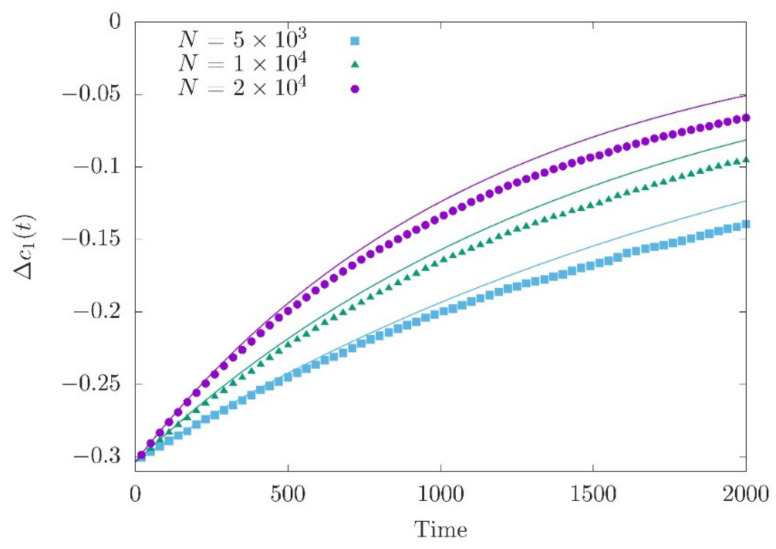
Approach of $\Delta c_{1}\left(t\right)$ to the new optimum for three different population sizes. Simulation data is compared with the theoretical expectation (solid curve) given by Equation (13). The new fitness optimum is $z_{f}=0.5.$ The mean deviation and genetic variance just before the optimum shift are $\Delta c_{1}\left(0\right)$ = (−0.304, −0.304, −0.303) and $c_{2}\left(0\right)$ = (0.0045, 0.0066, 0.0089) for *N* = 5000, 10,000 and 20,000, respectively. 200 independent loci are simulated with 200 iterations for averaging.

**Figure 2 entropy-22-00907-f002:**
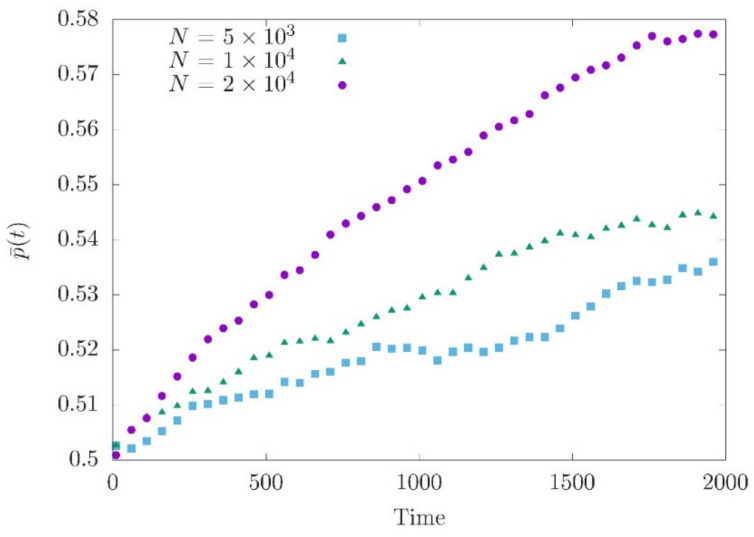
Average allele frequency $\bar{p}\left(t\right)$ after the optimum shift at loci with effect size around 0.01. A total of 200 independent loci are simulated. $\bar{p}\left(t\right)$ is obtained by averaging 200 simulation runs. This yields $\bar{p}\left(0\right)\approx 0.5$ for all loci.

**Figure 3 entropy-22-00907-f003:**
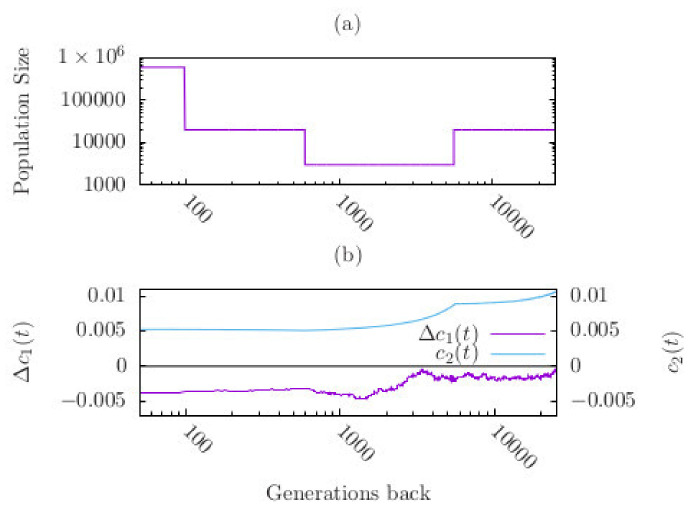
(**a**) Demography describing the change in effective population size for the past 25,600 generations (adapted from Schiffels & Durbin 2014 [43]). The bottleneck phase starts 5600 generations ago when the population size suddently decreased from the stationary value of $2\times 10^{4}$ to 3000 individuals. The bottleneck phase lasts for 5000 generations. After the recovery from the bottleneck (600 generations ago) population size is constant for 500 generations, before it dramatically increased 100 generations ago to its current size. (**b**) Mean and variance of a quantitative trait as a function of time. The curves were obtained by averaging 2000 simulation runs.

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
