# Peer review of "Polygenic Adaptation in a Population of Finite Size"

_entropy, 2020, doi:10.3390/e22080907_

Round 1
Reviewer 1 Report
This paper provides a perspective on current population-genetic understanding of allele-frequency dynamics during polygenic adaptation in populations of finite size. This topic is of interest, and the authors are well-positioned to write the piece. I enjoyed reading the review, though I did have some major concerns (below) along with a number of smaller concerns. This review is from the perspective of a geneticist who is not a regular reader of Entropy.
Major concerns:
--I am not completely clear on how this paper contrasts from John & Stephan (2020), which includes all the mathematical results and similar simulations. The authors are clear that much of this is a recapitulation of John & Stephan, but they are less clear about which parts are new. I think there is potential value in translating the results of John & Stephan to the audience of Entropy, but the choices made to advance that presentation are not obvious---the presentation seems to me similar to what it would be for an audience of quantitatively minded evolutionary geneticists, and I suspect the paper would be hard to follow without a background in evolutionary genetics.
--Adding to the previous point: Substantial pieces of the text are lifted verbatim from John & Stephan (2020). (I found several instances of several sentences each without doing a thorough search in the course of checking John & Stephan for more details on some points.) Of course, the authors are identical, and they are clear that they are drawing heavily on John & Stephan (2020), but I thought it was important to flag. I think It would be preferable for the authors to rephrase any blocks of a full sentence or longer pulled their previous work.
--There are some tensions involved in describing the simulated model as a model of human height. Overall I don’t think that the simulated results shown are particularly relevant to height in particular, but they are probably generally relevant to a lot of polygenic traits. Attempting to model height specifically (or any specific trait, really) is fraught at this point because of uncertainty about stratification and selection history. Height has been estimated to be influenced by several orders of magnitude more variants than included here (Boyle et al 2017 put it at ~10^5, I think). At the same time, this estimate is uncertain both by its nature and because of subtle stratification in the GIANT and Robinson et al GWAS data used. If anything near this number of variants is included, the assumption of free recombination used here becomes untenable. To get around stratification entirely, I did a quick check the sibling-based UKBB analysis in Berg et al. There are ~5000 loci w/ permutation p < .0001 when only ~1000 are expected under the null and an (incorrect) assumption of independence. (I couldn’t use genome-wide significance as a filter bc of the limited # of permutations performed.) I didn't check any LD etc; these could be localized and due to a smaller # of true causal variants. At the same time, this is an underpowered study with only a couple tens of thousands of sibling pairs and is probably an underestimate of associated variants.
Recent reports of stratification are mentioned to bolster the choice of 200 over 500 variants, but at the same time, Turchin et al is used to bolster claims that height has been under polygenic selection and to bolster the assumed selection strength used in simulation. In fact, the selection history proposed by Turchin is put into doubt by Berg et al and Sohail et al, I think much more seriously in doubt than the contention that many (>>500) variants affect height.
Taking this together, the tie of the simulated trait to height strikes me as confusing and unnecessary. I think that a review of the history of the last decade of height/selection research could be helpful in the paper, particularly if the model can clarify any of the observations. However, I think the simulation results might be stronger if it’s presented as just a hypothetical polygenic trait going through a bottleneck and subsequent recovery.
Minor points and typos:
-p. 1 The word “accustomed” in the first line strikes me as slightly odd. Maybe “suited,” “fit,” or “tailored”?
-p. 2 “demographic changes due to varying population size” -> demographic changes such as varying population size”
-p. 2 “population structure in combination with positive selection needs to be modeled to describe typical phenomena” I’m not quite sure what this is saying, would “positive selection needs to be modeled in combination with population structure to describe typical phenomena” be accurate?
-p. 2 “Yet, of course, it is not clear if adaptation can occur via such subtle changes in allele frequencies.” I think this sentence is either too vague or out of date. There are of course major open questions remaining in exactly how or when polygenic adaptation happens via subtle frequency shifts, but it is already clear from simulation studies that it *can* happen via subtle frequency shifts for certain parameters / trait architectures. (e.g. Hollinger et al 2018)
-p. 2 “The analysis of population genetic models of polygenic adaptation began very recently.” I agree that the recent work on which the authors focus and review is different from past work, but I feel this sentence should be more specific---why doesn't older work by Barton, Turelli, etc count, or say Bulmer 1972? Or, for that matter, why not Haldane (1930)? Also, I am not sure whether readers of Entropy will appreciate the distinction of quantitative vs. population genetics that’s being implicitly used here.
-p. 2 “their model is identical to ours” -> “their model is identical to the one we discuss below” (Given that this is a review, it is not clear to the reader yet that discussion will focus on a specific model)
-p. 2 “(See last paragraph)” Does this mean the final paragraph or the previous paragraph?
-p. 2 “before the shift the fitness optimum” -> “before the shift of the fitness optimum”
-p. 3, eq. 5, please spell out that u represents the + to – mutation rate and that v represents the – to + mutation rate.
-p. 4 What is the relevance for the results of mu*l << 1? (Please give a pointer about this in the text)
-p. 4 “We start our simulations with all loci having an equal number of + and – alleles.” This phrasing made me wonder whether I had misunderstood something and there were more than two alleles per site (or multiple sites in a nonrecombining locus). I think this means that p_i = .5 for all loci, and saying it that way might be clearer.
-p. 5 I found eq. 6 and 7 to be hard to follow as written. It’d be easier if the order of 6 and 7 were reversed, if it was noted in text that (6) is (5) with u = v, and if the first two terms in what is now (6) were reordered to match (5). Further, it’d help to see where (6) comes from if it were stated explicitly that
d (delta c) / dt = sum(gamma(dp/dt – dq/dt)) = sum(gamma(2dp/dt)).
-p. 5 If we are measuring in SD units, is the assumption of “small” effect sizes more or less an assumption that there are many involved loci and that their effect sizes are not too heterogeneous? Otherwise, might the negligibility of this term hinge on scaling of the phenotype?
-p. 3-5 Throughout the exposition of the math, I was confused about which mathematical statements were new (after reading, I think all of them appear in John & Stephan or in earlier Jain & Stephan papers?) It might help to put in equation numbers from previous papers to make these connections clearer (e.g. eq 12 here is eq 15 from John & Stephan).
-p. 5, I found myself confused about why there is a slight negative deviation from the optimum at equilibrium. Does this result from mutational pressure driving the mean phenotype down (perhaps because the mean p is greater than 50% at the optimum, but mutation is symmetric)? Whatever the explanation is, I think it would help to state it more explicitly.
-p. 5 As mentioned in the major point above, I think that the parameters used in the simulation are perfectly interesting but not especially convincing as a model for height specifically.
-p. 5 in the citation to Sella & Hirsh (2005) and in the references, Hirsh’s name is misspelled as “Hirsch”
-p. 6 and throughout, The description of the demography in Figure 3 as the “Human” demography is incomplete---it is only a representation of the demography of populations that experienced the out-of-Africa bottleneck (i.e. not for populations w/ much or all of their ancestry from sub-Saharan Africa, and not for populations that experienced additional bottlenecks after the OOA bottleneck). The authors may want to think about the history of height in Europe specifically, but the local relevance of the chosen parameters should be made clearer.
p.6 The first figure is labeled as Figure 3 (Figures 1 and 2 come later). In the figure caption, it would help to state explicitly that time is on a log scale.
-p. 6-10 I am confused about the relationship and ordering of the different simulations. It seems to me that Figure 3 (which appears first but is mostly discussed in section 3.3) does not include an optimum shift; instead it shows how closeness to the (fixed) optimum varies as demography changes. In contrast, Figures 1 and 2 seem to be about one set of simulations (distinct from those in Figure 3) in which an optimum shift is imposed but there is no demographic change. The relative ordering and discussion made this hard to follow (and I’m afraid I’m still understanding incorrectly).
-In the Figure 1 and 2 captions, please state the number of simulations and, if applicable, number of loci included in the summaries.
-p.7 “Furthermore, note that the drift term” : It might help to be explicit that this is a drift term in the drift-diffusion sense and not in the genetic drift sense.
-p. 9 The joint parameter that is emphasized right after eq. 19 is exactly what you’d expect from the single locus case, where speed of allele-frequency change is proportional to spq. It might be worth noting this.
-p. 9 For the discussion of the effect of allele frequency on genetic drift in the second paragraph, is this an accurate guide to intuition: selection-driven changes are proportional to p(1-p), whereas drift-driven changes are of order sqrt(p(1-p)) (since the variance due to drift is proportional to p(1-p)). So given an effect size, sel tends to dominate when p(1-p) / sqrt(p(1-p)) = sqrt(p(1-p)) is large, which happens for intermediate p.
-p. 9 “We started our simulation” -> “We started our simulations”
-p. 9 “This amounts to a about 43.76% decrease inΔc1 after the bottleneck” –three small things: First “a about” -> “an approximate.” Second, 43.76% seems a bit overprecise (same for the 42.74% a couple sentences later). Maybe ~44%? Third, “decrease” is a little confusing since the striking part (as I understand it) is that the absolute size is increasing. Maybe “decrease” -> “increase in the absolute size of”?
-p. 9 “This observation is obviously caused by genetic drift.” Instead of “obviously,” maybe explain that it’s clearly not caused by selection (which is constant) or mutation?
-p. 10 “We also analyzed the underlying allelic dynamics of the phenotypic response to selection. Whereas in a population of infinite size, the equilibrium frequency of small alleles is 0.5 (de Vladar & Barton 2014),” Does this also require an assumption of symmetric mutation? Also, should this be “small-effect alleles”?
Author Response
Our replies are in Arial/italic.
Comments and Suggestions for Authors
This paper provides a perspective on current population-genetic understanding of allele-frequency dynamics during polygenic adaptation in populations of finite size. This topic is of interest, and the authors are well-positioned to write the piece. I enjoyed reading the review, though I did have some major concerns (below) along with a number of smaller concerns. This review is from the perspective of a geneticist who is not a regular reader of Entropy.
Major concerns:
--I am not completely clear on how this paper contrasts from John & Stephan (2020), which includes all the mathematical results and similar simulations. The authors are clear that much of this is a recapitulation of John & Stephan, but they are less clear about which parts are new. I think there is potential value in translating the results of John & Stephan to the audience of Entropy, but the choices made to advance that presentation are not obvious---the presentation seems to me similar to what it would be for an audience of quantitatively minded evolutionary geneticists, and I suspect the paper would be hard to follow without a background in evolutionary genetics.
We now have clearly marked (in red) which parts of the paper have been changed due to similarity to John and Stephan (2020). It is impossible to list all these changes here, as they are distributed throughout the entire manuscript. Furthermore, we have made modifications to avoid discussions that are too specific for readers of this volume on statistical physics. In particular we have cut the tie between our bottleneck model and the evolution of human height (see below). And in the first two paragraphs of Introduction we have explained basic terms of evolutionary adaptation and evolutionary genetics.
--Adding to the previous point: Substantial pieces of the text are lifted verbatim from John & Stephan (2020). (I found several instances of several sentences each without doing a thorough search in the course of checking John & Stephan for more details on some points.) Of course, the authors are identical, and they are clear that they are drawing heavily on John & Stephan (2020), but I thought it was important to flag. I think It would be preferable for the authors to rephrase any blocks of a full sentence or longer pulled their previous work.
We removed longer passages that are highly similar between both papers (such as the last two paragraphs of the original ms on parameter values and on the description of the simulations are deleted (p. 4, para. 1), the paragraph after eq. (12) in the previous paper on the shape of the distribution (eq. 11) or the paragraph on the genetic variance at the end of the equilibrium analysis (p. 5)). Shorter stretches of identity (sentences) were reformulated as much as possible and are found throughout the manuscript (marked in red).
--There are some tensions involved in describing the simulated model as a model of human height. Overall I don’t think that the simulated results shown are particularly relevant to height in particular, but they are probably generally relevant to a lot of polygenic traits. Attempting to model height specifically (or any specific trait, really) is fraught at this point because of uncertainty about stratification and selection history. Height has been estimated to be influenced by several orders of magnitude more variants than included here (Boyle et al 2017 put it at ~10^5, I think). At the same time, this estimate is uncertain both by its nature and because of subtle stratification in the GIANT and Robinson et al GWAS data used. If anything near this number of variants is included, the assumption of free recombination used here becomes untenable. To get around stratification entirely, I did a quick check the sibling-based UKBB analysis in Berg et al. There are ~5000 loci w/ permutation p < .0001 when only ~1000 are expected under the null and an (incorrect) assumption of independence. (I couldn’t use genome-wide significance as a filter bc of the limited # of permutations performed.) I didn't check any LD etc; these could be localized and due to a smaller # of true causal variants. At the same time, this is an underpowered study with only a couple tens of thousands of sibling pairs and is probably an underestimate of associated variants.
Recent reports of stratification are mentioned to bolster the choice of 200 over 500 variants, but at the same time, Turchin et al is used to bolster claims that height has been under polygenic selection and to bolster the assumed selection strength used in simulation. In fact, the selection history proposed by Turchin is put into doubt by Berg et al and Sohail et al, I think much more seriously in doubt than the contention that many (>>500) variants affect height.
Taking this together, the tie of the simulated trait to height strikes me as confusing and unnecessary. I think that a review of the history of the last decade of height/selection research could be helpful in the paper, particularly if the model can clarify any of the observations. However, I think the simulation results might be stronger if it’s presented as just a hypothetical polygenic trait going through a bottleneck and subsequent recovery.
To improve readability of our paper (see reply to the first comment of this reviewer) and because of the difficulties mentioned by the reviewer with relating our demography model to human height, we followed his advice and presented our simulation results “as just a hypothetical polygenic trait going through a bottleneck and subsequent recovery” (p. 8, last three paragraphs). See also Abstract, p. 1, last four lines, and p. 3, second para., l. 5.
Minor points and typos:
-p. 1 The word “accustomed” in the first line strikes me as slightly odd. Maybe “suited,” “fit,” or “tailored”?
“accustomed” is replaced by “suited”, p. 1, last para., l. 1.
-p. 2 “demographic changes due to varying population size” -> demographic changes such as varying population size”
We now write “demographic changes such as varying population size”, p. 2, second para, l. 4.
-p. 2 “population structure in combination with positive selection needs to be modeled to describe typical phenomena” I’m not quite sure what this is saying, would “positive selection needs to be modeled in combination with population structure to describe typical phenomena” be accurate?
We now write “positive selection in combination with population structure..”, p. 2, second para, l. 5.
-p. 2 “Yet, of course, it is not clear if adaptation can occur via such subtle changes in allele frequencies.” I think this sentence is either too vague or out of date. There are of course major open questions remaining in exactly how or when polygenic adaptation happens via subtle frequency shifts, but it is already clear from simulation studies that it *can* happen via subtle frequency shifts for certain parameters / trait architectures. (e.g. Hollinger et al 2018)
We now write “These papers predicted that allele frequencies change by small amounts when a large number of genetic loci of minor effect sizes control a phenotypic trait.”, p. 2, para. 3, l. 4.
-p. 2 “The analysis of population genetic models of polygenic adaptation began very recently.” I agree that the recent work on which the authors focus and review is different from past work, but I feel this sentence should be more specific---why doesn't older work by Barton, Turelli, etc count, or say Bulmer 1972? Or, for that matter, why not Haldane (1930)? Also, I am not sure whether readers of Entropy will appreciate the distinction of quantitative vs. population genetics that’s being implicitly used here.
We now write “The analysis of models of polygenic adaptation has a long tradition…”, p. 2, para. 5, l. 1.
-p. 2 “their model is identical to ours” -> “their model is identical to the one we discuss below” (Given that this is a review, it is not clear to the reader yet that discussion will focus on a specific model)
We now write “… their model is identical to the one we discuss below”, p. 2, last para, l. 3 (from below).
-p. 2 “(See last paragraph)” Does this mean the final paragraph or the previous paragraph?
We now write “previous”, p. 3, first para., l. 5.
-p. 2 “before the shift the fitness optimum” -> “before the shift of the fitness optimum”
We now insert “of”, p. 3, second para., l. 2.
-p. 3, eq. 5, please spell out that u represents the + to – mutation rate and that v represents the – to + mutation rate.
We now write “Here, represents the mutation rate from the + to the allele and that of the to the + allele.“, p. 3, last para., l. -4.
-p. 4 What is the relevance for the results of mu*l << 1? (Please give a pointer about this in the text)
The paragraph on parameter values for the simulations (see p.4, paragraph 1) is deleted.
-p. 4 “We start our simulations with all loci having an equal number of + and – alleles.” This phrasing made me wonder whether I had misunderstood something and there were more than two alleles per site (or multiple sites in a nonrecombining locus). I think this means that p_i = .5 for all loci, and saying it that way might be clearer.
The paragraph on the description of the simulations (see p.4, paragraph 1) is deleted.
-p. 5 I found eq. 6 and 7 to be hard to follow as written. It’d be easier if the order of 6 and 7 were reversed, if it was noted in text that (6) is (5) with u = v, and if the first two terms in what is now (6) were reordered to match (5). Further, it’d help to see where (6) comes from if it were stated explicitly that
d (delta c) / dt = sum(gamma(dp/dt – dq/dt)) = sum(gamma(2dp/dt)).
We added the following sentence “Eq. (6) is derived by summing over the eqs. (5) and using the definitions of the cumulants (eqs. (1) to (3)).“ Eg. (6) is most important here and should stay first. Furthermore, the description how eq. (6) is derived shows that it comes from eqs. (5)., p. 4, para. 4, l. 1.
-p. 5 If we are measuring in SD units, is the assumption of “small” effect sizes more or less an assumption that there are many involved loci and that their effect sizes are not too heterogeneous? Otherwise, might the negligibility of this term hinge on scaling of the phenotype?
The definition of small and large effects is suitable for this model, as the qualitative behavior of allele frequencies depends critically on gamma. In general, however, this definition does not work, even in the case of uneven mutation rates.
-p. 3-5 Throughout the exposition of the math, I was confused about which mathematical statements were new (after reading, I think all of them appear in John & Stephan or in earlier Jain & Stephan papers?) It might help to put in equation numbers from previous papers to make these connections clearer (e.g. eq 12 here is eq 15 from John & Stephan).
We now included equation numbers from the previous paper when necessary (p. 4, para. 5, l. 4; p. 6, first paragraph below figure legend, l. 4).
-p. 5, I found myself confused about why there is a slight negative deviation from the optimum at equilibrium. Does this result from mutational pressure driving the mean phenotype down (perhaps because the mean p is greater than 50% at the optimum, but mutation is symmetric)? Whatever the explanation is, I think it would help to state it more explicitly.
We added a new paragraph explaining this, p. 4, paragraph 6 before eq. (9).
-p. 5 As mentioned in the major point above, I think that the parameters used in the simulation are perfectly interesting but not especially convincing as a model for height specifically.
We cut the relation of our demography simulations to human height, p. 5, para. 3, l. 2; p. 8, last three paragraphs.
-p. 5 in the citation to Sella & Hirsh (2005) and in the references, Hirsh’s name is misspelled as “Hirsch”
Citation deleted, p. 5, last para. of section 3.1.
-p. 6 and throughout, The description of the demography in Figure 3 as the “Human” demography is incomplete---it is only a representation of the demography of populations that experienced the out-of-Africa bottleneck (i.e. not for populations w/ much or all of their ancestry from sub-Saharan Africa, and not for populations that experienced additional bottlenecks after the OOA bottleneck). The authors may want to think about the history of height in Europe specifically, but the local relevance of the chosen parameters should be made clearer.
We cut the relation of our demography simulations to human height, p. 8, last three paragraphs.
p.6 The first figure is labeled as Figure 3 (Figures 1 and 2 come later). In the figure caption, it would help to state explicitly that time is on a log scale.
We reordered the figures. We think that it is clear that the time axis is on a log scale.
-p. 6-10 I am confused about the relationship and ordering of the different simulations. It seems to me that Figure 3 (which appears first but is mostly discussed in section 3.3) does not include an optimum shift; instead it shows how closeness to the (fixed) optimum varies as demography changes. In contrast, Figures 1 and 2 seem to be about one set of simulations (distinct from those in Figure 3) in which an optimum shift is imposed but there is no demographic change. The relative ordering and discussion made this hard to follow (and I’m afraid I’m still understanding incorrectly).
We do not know which version of our manuscript the reviewers saw. In our submitted version there was not a problem with the order of the figures. This problem has now been fixed.
-In the Figure 1 and 2 captions, please state the number of simulations and, if applicable, number of loci included in the summaries.
The number of simulations and loci are introduced in Figures 1 (p. 6) and 2 (p. 7).
-p.7 “Furthermore, note that the drift term” : It might help to be explicit that this is a drift term in the drift-diffusion sense and not in the genetic drift sense.
We now write “this drift term”. This is related to the “drift term” at the beginning of the paragraph, p. 6, second to last para, l. 4.
-p. 9 The joint parameter that is emphasized right after eq. 19 is exactly what you’d expect from the single locus case, where speed of allele-frequency change is proportional to spq. It might be worth noting this.
We now write “This is similar to the one-locus case, where the product of and determines the speed of allele frequency change.“, p. 8, first para, l. 2.
-p. 9 For the discussion of the effect of allele frequency on genetic drift in the second paragraph, is this an accurate guide to intuition: selection-driven changes are proportional to p(1-p), whereas drift-driven changes are of order sqrt(p(1-p)) (since the variance due to drift is proportional to p(1-p)). So given an effect size, sel tends to dominate when p(1-p) / sqrt(p(1-p)) = sqrt(p(1-p)) is large, which happens for intermediate p.
Yes.
-p. 9 “We started our simulation” -> “We started our simulations”
We now write “We started the simulations…”, p. 8, para. 4, l. 1.
-p. 9 “This amounts to a about 43.76% decrease inΔc1 after the bottleneck” –three small things: First “a about” -> “an approximate.” Second, 43.76% seems a bit overprecise (same for the 42.74% a couple sentences later). Maybe ~44%? Third, “decrease” is a little confusing since the striking part (as I understand it) is that the absolute size is increasing. Maybe “decrease” -> “increase in the absolute size of”?
We now write ”an approximate 41% increase in the absolute size of …”, p. 8, last para, l. 4.
-p. 9 “This observation is obviously caused by genetic drift.” Instead of “obviously,” maybe explain that it’s clearly not caused by selection (which is constant) or mutation?
We now write “This observation is caused by genetic drift.”, p. 8, last para., l.7.
-p. 10 “We also analyzed the underlying allelic dynamics of the phenotypic response to selection. Whereas in a population of infinite size, the equilibrium frequency of small alleles is 0.5 (de Vladar & Barton 2014),” Does this also require an assumption of symmetric mutation? Also, should this be “small-effect alleles”?
We now write this sentence as “Whereas in a population of infinite size, the equilibrium frequencies of small-effect alleles are intermediate (de Vladar & Barton 2014), we found that genetic drift (in conjunction with symmetric mutation and selection) may lead to a very different equilibrium distribution of allele frequencies.“, p. 9, last para, l. 1.
Reviewer 2 Report
The majority of the content of the current manuscript restates results (verbatim in many places) from a recently published study in E&E from the same authors. The only differences that I could discern are the addition of a new diffusion-based analysis (equations 14 to 19) that leads to an interesting on result on the impact of drift on rate of polygenic adaptation, and a discussion of relevant results from other theoretical studies of polygenic adaptation (which is largely preoccupied with results from the recent Hayward and Sella bioRxiv preprint).
Given that this manuscript is submitted as a review, too much time is spent recapitulating the results of the authors' recent E&E paper and insufficient attention is paid to reviewing the literature on polygenic adaptation. Indeed, a recent review in NRG (https://doi.org/10.1038/s41576-020-0250-z) provides a broad overview of the current state of the field, but lacks a detailed exposition of theoretical results in this area, a gap that could be bridged in a revised version of the current review.
Such a review would have broad appeal for those interested in ongoing developments and future directions of theoretical polygenic adaptation research, and I would encourage the authors to resubmit the current manuscript with this aim in mind.
Author Response
Comments and Suggestions for Authors
The majority of the content of the current manuscript restates results (verbatim in many places) from a recently published study in E&E from the same authors. The only differences that I could discern are the addition of a new diffusion-based analysis (equations 14 to 19) that leads to an interesting on result on the impact of drift on rate of polygenic adaptation, and a discussion of relevant results from other theoretical studies of polygenic adaptation (which is largely preoccupied with results from the recent Hayward and Sella bioRxiv preprint).
Given that this manuscript is submitted as a review, too much time is spent recapitulating the results of the authors' recent E&E paper and insufficient attention is paid to reviewing the literature on polygenic adaptation. Indeed, a recent review in NRG (https://doi.org/10.1038/s41576-020-0250-z) provides a broad overview of the current state of the field, but lacks a detailed exposition of theoretical results in this area, a gap that could be bridged in a revised version of the current review.
Such a review would have broad appeal for those interested in ongoing developments and future directions of theoretical polygenic adaptation research, and I would encourage the authors to resubmit the current manuscript with this aim in mind.
We have not intended to write a comprehensive review including both empirical and theoretical aspects of this field (see the statement in the second to last paragraph on p. 2, l. 2). We think that such a paper may not be appropriate for the readership of a statistical physics volume.
Reviewer 3 Report
John & Stephan present a nice summary of their previous work on models of polygenic adaptation. The paper is clear and easy to read, and on an important and interesting problem. I have mostly minor comments on a few specific passages that I thought could be improved. My only “major” comment is that, if this is really a “review” paper (as stated on the header), it focuses a bit heavily on the specific models that have been studied by the authors. While they have drawn parallels with work from other groups, they haven’t taken the extra step of showing exactly where the assumptions or analyses differ. As such, I think a comment on the limitations of scope of the review should be noted somewhere in the introduction, to indicate that this is not a comprehensive review of research on this topic.
General minor comment: In the author’s model, the selection strength is controlled by a parameter s which appears in a Gaussian stabilizing selection function. The author's results seem to implicitly assume that 1/s is much larger than the phenotypic variance. As far as I can tell, this condition seems to be met in the simulations, since they report genetic variance on the order of 0.005 for various simulations but use s=0.1. Do the authors think this condition will be met for real traits, in which the number of causal loci could potentially be very large? Or in very large populations, which (may) have more genetic variation?
General minor comment: The authors do not discuss heritability in the paper – their simulated traits seem to be perfectly heritable. This is not generally true of real traits. For a given genetic variance, will a highly heritable trait adapt faster than a less heritable trait? If a trait has low heritability but large phenotypic variance relative to the selection strength, this could affect the per-site selection strength, as noted above. Some discussion of the role of h^2 would be helpful.
General minor comment: The authors simulate 200 causal loci and a mutation rate (10^-5) that is orders of magnitude higher than the rate in humans. If the true rate per base in humans is order of 10^-8 as assumed in most papers, is this simulation roughly equivalent to 20000 loci (i.e., 10^3 as many as are actually simulated) at the 10^-8 rate?
I don’t necessarily have a problem with the author’s choice -- it is standard in many population genetic analyses to rescale mutation rates for computational efficiency, but since the authors are making a claim that this simulation is relevant to human height some note on the choice/relevance of this parameter should be given. It seems to me that only a fraction of these 200 sites will have large effect, high frequency alleles in any given simulation output, meaning that there must be far fewer effective causal sites in their simulations than there are true causal sites for height in the human genome.
(Below I will generally reference a quote by the authors and then note where I think these statements could be clarified or modified).
Page 2:
The authors write “An opinion paper entitled “Adaptation – not by sweeps alone” by Pritchard and Di Rienzo (2010) drew the attention of population geneticists to this type of selection.” I think this is basically true, but Pritchard, Pickrell, and Coop (Current Biology) also raised this issue and was published earlier in 2010.
“Yet, of course, it is not clear if adaptation can occur via such subtle changes in allele frequencies” I would argue that theory makes it clear that adaptation CAN occur by subtle allele frequency changes, but it is empirically unclear whether/when it actually does.
“The analysis of population genetic models of polygenic adaptation began very recently.” Are the Lande papers cited by the authors from the 1970s not in essence polygenic adaptation papers? Or perhaps Burger & Lynch 1995 (Evolution)?
Page 3-4:
“Human height is controlled by more than 500 loci (Wood et al. 2014), although recent estimates suggest that this number is probably too high since population structure has not been adequately 4 considered in GWAS (Berg et al. 2019; Sohail et al. 2019).” I don’t quite agree with this sentence. The 500 number is not the number of “loci” that affect the trait, it is the number of (roughly independent) high-frequency, moderate-effect alleles that were detected in GWAS. Alleles that are too rare or too weak are not included since they cannot be detected in a standard GWAS. Additionally, while it is certainly possible that some previous hits are false positives due to stratification, the UKBB GWAS finds highly correlated (and almost always same-direction effects) as the stratified GIANT study (See figure 3A of Berg et al 2019).
Page 4:
“for 2N generations such that the allele frequencies stabilize” perhaps replace stabilize with equilibrate?
Page 6:
“Using the simulation data from John and Stephan (2020) we find a reasonable fit of the theoretical predictions with the simulation results averaged over 50000 independent runs.” Are the authors referring to the data plotted in figure 3? If so perhaps the theoretical expectation could be added to this plot, and if not perhaps they can point the readers to a relevant figure?
Page 8:
“We can clearly see that the smallest population shows only a minor directional change in allele frequency indicating a stronger impact of genetic drift (Figure 2).” While I’m sure the authors are correct in their overall conclusion here, whether the changes are directional is not entirely obvious from the plot. Since the authors have initialized the allele frequencies at 0.5 for this, any deviation away from p=q=0.5 will result in reduction of p*(1-p), whether due to directional selection or drift alone.
Page 9:
“This appears to be an important property of polygenic selection because it may help detecting this type of selection, although the frequency shifts at individual loci are in general small (discussed in Stephan (2016) and Jain & Stephan (2017b)).” I find the self-citation a bit odd here – many earlier papers used this qualitative result to look for empirical signals of polygenic adaptation, including several the authors have cited elsewhere in the manuscript.
“Second, however, for the allele frequencies we get a reasonable agreement of the deterministic prediction of eqs. (19) and simulations only when the effect sizes are sufficiently large and allele frequencies at the time of the environmental shift are intermediate”. Is this agreement intended to be plotted anywhere? There are only simulation results in figure 2, not theoretical expectations.
“This observation is obviously caused by genetic drift” I would suggest removing the word “obviously” as it is often discouraging to readers who are new to the field.
Lastly, at least one passage in the paper appears to be identical to passages in the author’s recent Ecology & Evolution paper on a similar topic. “Theories with very different assumptions about mutation (Lande’s (1976) model with no explicit loci, Barton’s (1989) mutation-selection-drift model similar to ours and Sella and Hirsch’s (2005) weak-mutation Markov chain approximation), all predict that the stationary distribution of the mean deviation from the optimum should have variance 1/(2Ns). This is a quite generic property of stochastic processes best known for the OrnsteinUhlenbeck process (Simons et al. 2018)” appears in both papers. While generally I am not averse to authors repackaging their own work for new audiences, it seems that this level of redundancy is quite high, especially given that the two papers have essentially the same modeling and even discuss some of the same simulations results. I would suggest the authors comb through the manuscript to make sure that such identical phrases are removed.
Author Response
John & Stephan present a nice summary of their previous work on models of polygenic adaptation. The paper is clear and easy to read, and on an important and interesting problem. I have mostly minor comments on a few specific passages that I thought could be improved. My only “major” comment is that, if this is really a “review” paper (as stated on the header), it focuses a bit heavily on the specific models that have been studied by the authors. While they have drawn parallels with work from other groups, they haven’t taken the extra step of showing exactly where the assumptions or analyses differ. As such, I think a comment on the limitations of scope of the review should be noted somewhere in the introduction, to indicate that this is not a comprehensive review of research on this topic.
We have followed the advice of the referee and added a comment (see the second to last paragraph of p. 2, l.1) that our paper is not intended to be a classical review (i.e. comprehensive) on this field.
General minor comment: In the author’s model, the selection strength is controlled by a parameter s which appears in a Gaussian stabilizing selection function. The author's results seem to implicitly assume that 1/s is much larger than the phenotypic variance. As far as I can tell, this condition seems to be met in the simulations, since they report genetic variance on the order of 0.005 for various simulations but use s=0.1. Do the authors think this condition will be met for real traits, in which the number of causal loci could potentially be very large? Or in very large populations, which (may) have more genetic variation?
We have added a sentence that 1/s is assumed to be much larger than the phenotypic variance (paragraph after eq. (4) on p. 3, l.1). Regarding the parameter values we followed here Turchin et al. 2012 and especially Simons et al. 2018. The latter studied several human traits analyzed by GWAS data and are probably the best source for choosing relevant values of parameters for quantitative traits (cited in last paragraph of section 3.1 on p. 5, l. 3). Furthermore, they analyzed a model in Simons et al. that is closely related to ours (cited in last para. on p. 2, l. 3).
General minor comment: The authors do not discuss heritability in the paper – their simulated traits seem to be perfectly heritable. This is not generally true of real traits. For a given genetic variance, will a highly heritable trait adapt faster than a less heritable trait? If a trait has low heritability but large phenotypic variance relative to the selection strength, this could affect the per-site selection strength, as noted above. Some discussion of the role of h^2 would be helpful.
We now mention this (paragraph 3 on p. 3, l. 3). In all papers since de Vladar and Barton (2014) heritability is not explicitly modeled. For this reason we have not considered it here, although it is certainly relevant and should be studied.
General minor comment: The authors simulate 200 causal loci and a mutation rate (10^-5) that is orders of magnitude higher than the rate in humans. If the true rate per base in humans is order of 10^-8 as assumed in most papers, is this simulation roughly equivalent to 20000 loci (i.e., 10^3 as many as are actually simulated) at the 10^-8 rate?
I don’t necessarily have a problem with the author’s choice -- it is standard in many population genetic analyses to rescale mutation rates for computational efficiency, but since the authors are making a claim that this simulation is relevant to human height some note on the choice/relevance of this parameter should be given. It seems to me that only a fraction of these 200 sites will have large effect, high frequency alleles in any given simulation output, meaning that there must be far fewer effective causal sites in their simulations than there are true causal sites for height in the human genome.
The estimate of the mutation rate is one of the parameter values we found in the literature on human traits. To the best of our knowledge, it is reliable (see Simons et al. 2018). As mentioned in our reply to Referee 1, we do no longer relate our simulation results to human height (see the demography section 3.3 on p. 8).
(Below I will generally reference a quote by the authors and then note where I think these statements could be clarified or modified).
Page 2:
The authors write “An opinion paper entitled “Adaptation – not by sweeps alone” by Pritchard and Di Rienzo (2010) drew the attention of population geneticists to this type of selection.” I think this is basically true, but Pritchard, Pickrell, and Coop (Current Biology) also raised this issue and was published earlier in 2010.
We now include Pritchard et al. (2010) on p. 2, para. 3, l. 3.
“Yet, of course, it is not clear if adaptation can occur via such subtle changes in allele frequencies” I would argue that theory makes it clear that adaptation CAN occur by subtle allele frequency changes, but it is empirically unclear whether/when it actually does.
We now mention this on p. 2, para. 3, l. 4.
“The analysis of population genetic models of polygenic adaptation began very recently.” Are the Lande papers cited by the authors from the 1970s not in essence polygenic adaptation papers? Or perhaps Burger & Lynch 1995 (Evolution)?
Both references are included on p. 2, para. 5, l. 1.
Page 3-4:
“Human height is controlled by more than 500 loci (Wood et al. 2014), although recent estimates suggest that this number is probably too high since population structure has not been adequately 4 considered in GWAS (Berg et al. 2019; Sohail et al. 2019).” I don’t quite agree with this sentence. The 500 number is not the number of “loci” that affect the trait, it is the number of (roughly independent) high-frequency, moderate-effect alleles that were detected in GWAS. Alleles that are too rare or too weak are not included since they cannot be detected in a standard GWAS. Additionally, while it is certainly possible that some previous hits are false positives due to stratification, the UKBB GWAS finds highly correlated (and almost always same-direction effects) as the stratified GIANT study (See figure 3A of Berg et al 2019).
We have decided to avoid this discussion in this review. Instead, we follow the suggestion of Rev. 1 and present our simulation results “as just a hypothetical polygenic trait going through a bottleneck and subsequent recovery” (p. 8, Section 3.3).
Page 4:
“for 2N generations such that the allele frequencies stabilize” perhaps replace stabilize with equilibrate?
We deleted the paragraph describing the simulations (see section 2.2 on p. 4).
Page 6:
“Using the simulation data from John and Stephan (2020) we find a reasonable fit of the theoretical predictions with the simulation results averaged over 50000 independent runs.” Are the authors referring to the data plotted in figure 3? If so perhaps the theoretical expectation could be added to this plot, and if not perhaps they can point the readers to a relevant figure?
This paragraph (last one of section 3.1 on p. 5) is rewritten. Eq. (8) is now compared with data from our demography simulation (Figure 3), i.e. from the equilibrium phase before the bottleneck.
Page 8:
“We can clearly see that the smallest population shows only a minor directional change in allele frequency indicating a stronger impact of genetic drift (Figure 2).” While I’m sure the authors are correct in their overall conclusion here, whether the changes are directional is not entirely obvious from the plot. Since the authors have initialized the allele frequencies at 0.5 for this, any deviation away from p=q=0.5 will result in reduction of p*(1-p), whether due to directional selection or drift alone.
We changed Figure 2 by plotting the average allele frequency against time. This shows better that the changes are directional. See Figure 2 on p. 7 and the paragraph (l. 4) above this figure.
Page 9:
“This appears to be an important property of polygenic selection because it may help detecting this type of selection, although the frequency shifts at individual loci are in general small (discussed in Stephan (2016) and Jain & Stephan (2017b)).” I find the self-citation a bit odd here – many earlier papers used this qualitative result to look for empirical signals of polygenic adaptation, including several the authors have cited elsewhere in the manuscript.
The self-citations have been deleted (first para., p. 8, l. 7).
“Second, however, for the allele frequencies we get a reasonable agreement of the deterministic prediction of eqs. (19) and simulations only when the effect sizes are sufficiently large and allele frequencies at the time of the environmental shift are intermediate”. Is this agreement intended to be plotted anywhere? There are only simulation results in figure 2, not theoretical expectations.
This was plotted in Figure 5 of the previous paper John and Stephan (2020).
“This observation is obviously caused by genetic drift” I would suggest removing the word “obviously” as it is often discouraging to readers who are new to the field.
“obviously” is now deleted (p. 8, last para., l. 7).
Lastly, at least one passage in the paper appears to be identical to passages in the author’s recent Ecology & Evolution paper on a similar topic. “Theories with very different assumptions about mutation (Lande’s (1976) model with no explicit loci, Barton’s (1989) mutation-selection-drift model similar to ours and Sella and Hirsch’s (2005) weak-mutation Markov chain approximation), all predict that the stationary distribution of the mean deviation from the optimum should have variance 1/(2Ns). This is a quite generic property of stochastic processes best known for the OrnsteinUhlenbeck process (Simons et al. 2018)” appears in both papers. While generally I am not averse to authors repackaging their own work for new audiences, it seems that this level of redundancy is quite high, especially given that the two papers have essentially the same modeling and even discuss some of the same simulations results. I would suggest the authors comb through the manuscript to make sure that such identical phrases are removed.
We removed longer passages that are highly similar between both papers (such as the last two paragraphs of the original ms on parameter values and on the description of the simulations are deleted (p. 4, para. 1), the paragraph after eq. (12) in the previous paper on the shape of the distribution (eq. 11) or the paragraph on the genetic variance at the end of the equilibrium analysis (p. 5)), which the referee refers to above.
Round 2
Reviewer 1 Report
The authors have responded to my concerns. At this point I have no major new points to raise and defer to the editors' judgment about the paper's fit to the journal, of which I am not a regular reader.